# Clinical significance of hypoalbuminemia in patients with scrub typhus complicated by acute kidney injury

**Ju Hwan Oh**, **Ji Hye Lim, A. Young Cho, Kwang Young Lee, In O. Sun***

Division of Nephrology, Department of Internal Medicine, Presbyterian Medical Center, Jeonju, Korea

* inogood@catholic.ac.kr

## Abstract

### Background

This study aimed to investigate the clinical significance of hypoalbuminemia (HA) in patients with scrub typhus complicated by acute kidney injury (AKI).

### Methods

From 2009 to 2018, 611 patients were diagnosed with scrub typhus. We divided the patients into two groups [normoalbuminemia (NA) vs. HA] based on the serum albumin level of 3.0 g/dL and compared the incidence, clinical characteristics, and severity of AKI based on the RIFLE classification between the two groups.

### Results

Of the total 611 patients, 78 (12.8%) were categorized into the HA group. Compared to patients in the NA group, patients in the HA group were older (73 ± 9 vs. 62 ± 14 years, P<0.001). The HA group had a significantly longer hospital stay (9.6 ± 6.2 vs 6.2 ± 3.1 days, p<0.001) and a higher incidence of complications in respiratory and cardiovascular systems. Furthermore, AKI developed significantly more in patients in the HA group (58% vs. 18%, p<0.001) as compared to the NA group. The overall incidence of AKI was 23.1%; of which, 14.9%, 7.0%, and 1.2% of cases were classified as Risk, Injury, and Failure, respectively. The serum albumin level correlated with AKI severity (3.4 ± 0.5 vs 3.0 ± 0.5 vs 2.6 ± 0.3, p<0.05). In a multivariate logistic regression analysis for predicting AKI, age, presence of co-morbidities such as chronic kidney disease, total bilirubin, leukocytosis, and hypoalbuminemia were significant predictors for AKI.

### Conclusion

Serum albumin level is helpful to predict the development and severity of scrub typhus-associated AKI.

**Data Availability Statement:** All relevant data are within the manuscript and its Supporting Information files.

**Funding:** Unfunded study The authors received no specific funding for this work.

**Competing interests:** The authors have declared that no competing interests exist.

**Abbreviations:** HA, hypoalbuminemia; AKI, acute kidney injury; NA, normoalbuminemia; AST, aspartate aminotransferase; ALT, alanine aminotransferase; DM, diabetes mellitus; CKD, chronic kidney disease; eGFR, estimated glomerular filtration rate; MDRD, Modification of Diet in Renal Disease; FENa, fractional excretion of sodium; ICU, intensive care unit.

## Introduction

Scrub typhus, caused by infection with *Orientia tsutsugamushi*, is a major cause of febrile illness in Southeast Asia [1]. The clinical spectrum of scrub typhus is wide, ranging from mild and probably subclinical to severe and fatal [2–4]. Severe complications of scrub typhus include pneumonitis, acute respiratory distress syndrome, meningitis, myocarditis, acute kidney injury (AKI), or even multiple organ failure, which may lead to death [4–6]. Thus, it becomes essential for clinicians to detect complications during the initial presentation.

The incidence of AKI among scrub typhus patients varies from 21 to 43% [7–9], and comorbidities, and biomarkers such as serum neutrophil gelatinase-associated lipocalin, are regarded as predictors for AKI in patients with scrub typhus [9, 10]. Hypoalbuminemia (HA) is known to be associated with complications and mortality in patients with acute infectious diseases [11] and has been correlated with an increased risk of AKI occurrence [12, 13]. Previous studies have shown that HA in scrub typhus was closely related to the frequency of various complications including AKI [14, 15]. However, the sample size of previous studies was relatively small, and there was no literature on whether hypoalbuminemia correlated with AKI severity based on the RIFLE criteria.

Therefore, we investigated the clinical significance of hypoalbuminemia in patients with scrub typhus complicated by AKI.

## Methods

### Patient selection

Between 2009 and 2018, we reviewed 628 patients with an acute febrile illness and a rash who were diagnosed with scrub typhus confirmed by a positive IgM ELISA (enzyme-linked immunosorbent assay) (InBios International Inc., Seattle, WA) for scrub typhus. Patients who were transferred to another hospital for a higher level of care, or to be treated nearer to their own homes during the course of treatment or had concomitant infections like leptospirosis, malaria, or dengue fever were excluded from the study. In addition, patients who were not admitted to our hospital were excluded from this study. We also excluded patients who were not followed-up during the complete recovery in their renal function or for at least three months after discharge. However, we included patients who were transferred from a local hospital after their baseline renal function had been checked. Therefore, a total of 611 patients were enrolled in this study and were divided into two groups [normoalbuminemia (NA) vs. HA] based on the serum albumin level of 3.0 g/dL. This study was approved by the Institutional Review Board of the Presbyterian Medical Center, Jeonju, South Korea (IRB No. 2020-04-017).

### Clinical and laboratory information

The patients' baseline demographic, clinical, and laboratory data were reviewed at the time of hospitalization and during the follow-up period. Detailed clinical histories of all patients were determined and they underwent thorough physical and biochemical examinations. A standard set of investigations including the following were carried out: complete blood counts, liver function and renal function tests, chest radiograph, three peripheral blood smears for malaria, urinalysis (including urea and electrolytes), two blood cultures, and a standard set of febrile serological investigations including an ELISA for the detection of scrub typhus (InBios International Inc., Seattle, WA). To differentiate between endemic zoonoses including leptospirosis and hemorrhagic fever with renal syndrome (HFRS), another serological work-up was performed. Leptospirosis was diagnosed by a $\geq$ 4-fold rise in indirect immunofluorescent assay

(IFA) titer in paired serum samples, or an antibody titer ≥ 1:800 in one serum sample by the microscopic agglutination test. HFRS was diagnosed by a single titer ≥1:80 or ≥ 4-fold rise in IFA titer in paired serum samples.

The complications were defined based on the presence of dysfunction of the following organ systems: 1) Cardiovascular system—presence of any of the following: systolic blood pressure <90 mmHg, myocarditis (defined using the diagnostic criteria for clinically suspected myocarditis presented by the European Society of Cardiology [16]), or new-onset cardiac arrhythmia including atrial fibrillation or supraventricular tachycardia; 2) Respiratory system —presence of any of the following: acute respiratory failure (defined not only by the incidence of hypoxemia, hypercapnia, or the failure to maintain both within a normal range, but also by the presence of any condition that resulted in the need for mechanical ventilation), pneumonia, pleural effusion, pulmonary edema, focal atelectasis, patchy consolidation on chest radiograph or acute distress syndrome; 3) Central nervous system—presence of any of the following: Glasgow Coma Scale ≤12 in the absence of any underlying causes, seizure without underlying causes, meningitis (defined by the presence of headache or nuchal rigidity with either altered sensorium or focal neurological deficits on history or examination, and with cerebrospinal fluid cell counts ≥5 leukocytes/mm$^3$); 4) Urinary system—presence of AKI, defined based on the RIFLE (Risk, Injury, Failure, Loss of kidney function, and End-stage kidney disease) criteria; 5) Gastrointestinal and hepatobiliary system—presence of any of the following: presence of hepatitis (defined as an elevation of serum transaminases more than five times the normal upper limit), or hyperbilirubinemia as serum total bilirubin >2 mg/dL, or gastrointestinal bleeding [9, 17–19].

Patients with AKI were categorized into the Risk (R), Injury (I) and, Failure (F) [20]. The estimated glomerular filtration rate (eGFR) was estimated using the abbreviated Modification of Diet in Renal Disease (MDRD) equation [21]. When the baseline renal function was not obtainable, it was calculated using the standard four-variable MDRD formula, assuming an eGFR of 75 mL/min/1.73 m$^2$. The RIFLE class was determined according to the worst of either the serum creatinine, or eGFR, and urine output criteria. Renal replacement therapy including intermittent hemodialysis was initiated using standard indications. All data are presented as mean ± standard deviation unless otherwise specified. The baseline characteristics of patients in the two groups were compared using $t$-tests, the chi-square test, or Fisher's exact test, as appropriate. The clinically relevant parameters or the variables that were significantly associated with the presence of AKI in the univariate analysis were included in the multivariate analysis. The AKI group was further grouped into three subgroups, then serum albumin level was compared among the three subgroups by the one-way analysis of variance followed by post Hoc analysis. A p-value of <0.05 was considered statistically significant. Statistical analysis was carried out using SPSS version 22.0 (IBM Corp., Armonk, NY).

## Results

### Comparison of clinical characteristics between the NA and HA groups

Compared to the patients in NA group, the patients in the HA group were older (73 ± 9 vs. 62 ± 14 years, P<0.001) and had a higher total leukocyte count (9.6 × 10$^3$/ mL vs. 6.2 × 10$^3$/ mL, P<0.001) (Table 1).

The duration of the hospital stay was longer for patients in HA group than NA group (9.6 ± 6.2 vs 6.2 ± 3.1 days, p<0.001). Furthermore, the patients in HA group required ICU care more frequently than those in NA group (20.5% vs 2.1%, P<0.001). The HA group also had a higher incidence of complications in the respiratory (50% vs 14%, P<0.001), cardiovascular (28% vs 11%, P<0.001), and neurologic systems (6.4% vs 0.8%, P = 0.002). In addition,

**Table 1. Comparison of baseline characteristics between NA and HA groups.**

| | NA | HA | P-value |
|---|---|---|---|
| | (n = 533) | (n = 78) | |
| Age | 62 ± 14 | 73 ± 9 | < 0.001 |
| Male, n(%) | 241 (45) | 30 (39) | NS |
| Duration of hospital stay, days | 6.2 ± 3.1 | 9.6 ± 6.2 | < 0.001 |
| DM, n(%) | 90 (17) | 19 (24) | NS |
| Hypertension, n(%) | 188 (35) | 40 (51) | 0.005 |
| CKD, n(%) | 19 (3) | 3 (4) | NS |
| ICU care, n(%) | 11 (2.1) | 16 (20.5) | < 0.001 |
| Hemoglobin (mg/dl) | 12.8 ± 1.6 | 11.5 ± 1.6 | < 0.001 |
| Complications | | | |
| Gastrointestinal tract, n (%) | 68 (13) | 8 (10) | NS |
| Respiratory system | 73 (14) | 39 (50) | < 0.001 |
| Cardiovascular system | 58 (11) | 22 (28) | < 0.001 |
| Central nervous system | 4 (0.8) | 5 (6.4) | 0.002 |
| Acute kidney injury | 96 (18) | 45 (58) | < 0.001 |
| Leukocyte ($\times 10^3$/ mL) | 6.7 ± 4.1 | 10.2 ± 3.8 | < 0.001 |
| Platelet count ($\times 10^3$/ mL) | 141± 57 | 122 ± 48 | 0.006 |
| Total bilirubin level | 0.7 ± 0.4 | 0.7 ± 0.7 | NS |
| Serum albumin (mg/dl) | 3.7 ± 0.4 | 2.7 ± 0.2 | < 0.001 |
| Serum ALT (IU/L) | 91 ± 129 | 76 ± 56 | NS |
| Creatinine (mg/dl) | 1.0 ± 0.4 | 1.3 ± 0.8 | < 0.001 |
| eGFR ml/min/1.73m$^2$ | 71 ± 24 | 50 ± 25 | < 0.001 |

AKI (58% vs 18%, P<0.01) developed more frequently in HA group than in NA group. The plasma ALT concentrations did not differ between the two groups, whereas the eGFR on admission (50 ± 25 ml/min/1.73m$^2$ vs. 71 ± 24 ml/min/1.73m$^2$, P<0.001) was lower in the HA group than in NA group.

## Comparison of clinical characteristics between the non-AKI group and AKI group

Compared to patients in the non-AKI group, the patients in the AKI group were older (71 ± 11 vs. 62 ± 14 years, P<0.001) and had a higher incidence of comorbidities, such as hypertension, diabetes, or chronic kidney disease (Table 2).

In addition, patients in the AKI group had poorer renal function (39 ± 17 vs. 78 ± 20 mL/min/1.73m$^2$, P<0.01) on admission and had a higher total leukocyte count (10.01 × 10$^3$/ mL vs. 6.78 × 10$^3$/mL, P<0.01). The plasma ALT concentrations did not differ between the two groups. However, the serum albumin level (3.2 ± 0.5 mg/dL vs. 3.7 ± 0.5 mg/dL, P<0.001) was lower in the AKI group than that of the non-AKI group.

## The relationship between serum albumin and acute kidney injury

Among patients with AKI, 15 (11%) were oliguric (Table 3).

One hundred and thirty-five patients had AKI prior to admission and 6 patients experienced AKI during their hospitalization. Of the 141 patients with AKI, 133 experienced a return to their baseline renal function within 72 hours after admission. Of the 83 patients with available fractional excretion of sodium (FENa) data, 53 (64%) had a FENa < 1%. Applying the

**Table 2. Comparison of baseline characteristics between AKI and non-AKI group.**

|  | Non-AKI | AKI | P-value |
|---|---|---|---|
|  | (n = 470) | (n = 141) |  |
| Age | 62 ± 14 | 71 ± 11 | < 0.001 |
| Male, n(%) | 200 (43) | 71 (50) | NS |
| Duration of hospital stay, days | 6.1 ± 2.7 | 8.4 ± 5.7 | < 0.001 |
| DM, n(%) | 66 (14) | 43 (31) | < 0.001 |
| Hypertension, n(%) | 138 (29) | 90 (64) | < 0.001 |
| CKD, n(%) | 5 (1) | 17 (12) | < 0.001 |
| Hemoglobin (mg/dl) | 12.9 ± 1.6 | 12.1 ± 1.9 | < 0.001 |
| Leukocyte (×10³/ mL) | 6.5 ± 4.0 | 9.3 ± 4.3 | < 0.001 |
| Platelet count (×10³/ mL) | 143 ± 57 | 121 ± 53 | < 0.001 |
| Total bilirubin level (mg/dl) | 0.7 ± 0.3 | 0.8 ± 0.6 | NS |
| Serum albumin (mg/dl) | 3.7 ± 0.5 | 3.2 ± 0.5 | < 0.001 |
| Serum ALT (IU/L) | 90 ± 119 | 83 ± 133 | NS |
| Creatinine (mg/dl) | 0.8 ± 0.2 | 1.5 ± 0.7 | < 0.001 |
| eGFRml/min/1.73m$^2$ | 78 ± 20 | 39 ± 17 | < 0.001 |

RIFLE criteria, 91 (64.5%), 43 (30.5%), and 7 (5.0%) patients were classified into the R, I, and F categories, respectively. The mean serum albumin concentrations differed among categories (Risk: 3.4 ± 0.5 mg/dL vs. Injury: 3.0 ± 0.5 mg/dL vs. Failure: 2.6 ± 0.3 mg/dL, P<0.005) (Fig 1).

On univariate analysis, age, presence of comorbidities, such as hypertension, diabetes, or chronic kidney disease, hemoglobin, thrombocytopenia, total leukocyte count, and hypoalbuminemia were significant predictors of AKI. After adjusting for these factors in multivariate logistic regression analysis, the presence of comorbidities, leukocytosis, and hypoalbuminemia were significant predictors of AKI (Table 4).

**Table 3. Clinical characteristics of 141 patients with AKI.**

| | |
|---|---|
| **Non-oliguric** | 15 (11) |
| **FENa < 1%, n (%)[a]** | 53 (64) |
| **Recovery of renal function within 72 h, n (%)** | 138 (98) |
| **Renal function** | |
| eGFR adm, ml/min/1.73m$^2$ | 39 ± 17 |
| eGFR low, ml/min/1.73m$^2$ | 36 ± 16 |
| eGFR rec, ml/min/1.73m$^2$ | 74 ± 21 |
| **Urinalysis results** | |
| Proteinuria | 15 (32) |
| Pyuria | 5 (10) |
| Hematuria | 1 (2) |
| Normal | 26 (56) |

FENa: fractional excretion of sodium

[a] FENa was available in 83 patients.

eGFR adm, ml/min/1.73m$^{2:}$ eGFR at the time of admission

eGFR low, ml/min/1.73m$^{2:}$ eGFR at lowest value in hospital

eGFR rec, ml/min/1.73m$^{2:}$ eGFR at the time of recovery

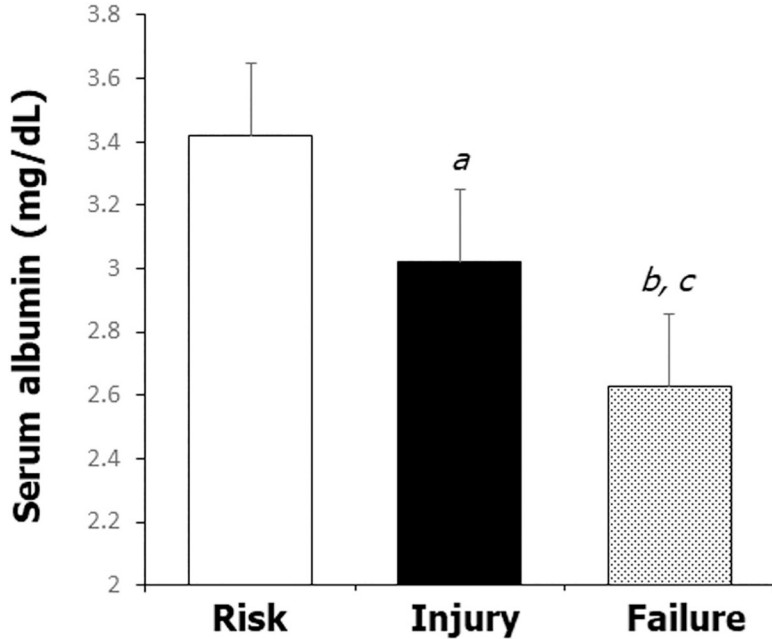

**Fig 1. The mean serum albumin concentrations among categories in patients with AKI.** The serum albumin level correlated with AKI severity. [a]$p<0.05$ vs. Risk, [b]$p<0.05$ vs. Risk, [c]$p<0.05$ vs. Injury.

## Discussion

Patients with HA showed a higher incidence of scrub typhus-associated complications including AKI in this study. Furthermore, HA was a significant predictor for scrub typhus-associated AKI and correlated with AKI severity. Therefore, our findings provide a rationale for close monitoring and aggressive therapy in patients with scrub typhus and HA.

In general, HA is associated with complications and mortality in patients with acute infectious diseases. HA is also frequently observed in scrub typhus with a rate of 25–69% [14, 22], and the association between HA and scrub typhus-related complications have been reported in previous studies [14, 15]. These findings were also observed in our study, where patients with HA experienced various complications more frequently in comparison to the NA group. Compared to previous studies, ours has the advantage of having enrolled in a larger sample. Therefore, the relationship between HA and scrub typhus-related complications was demonstrated in our study using a large cohort. In this study, the incidence of hypoalbuminemia was

**Table 4. Predictors of AKI (univariative and multivariative analysis).**

|  | Univariate | | Multivariate | |
|---|---|---|---|---|
|  | **HR (95% CI)** | **p-value** | **HR (95% CI)** | **p-value** |
| Age | 1.054 (1.036–1.072) | <0.001 | 1.019 (0.999–1.040) | 0.064 |
| DM | 2.686 (1.725–4.183) | <0.001 | 2.037 (1.179–3.519) | 0.011 |
| Hypertension | 4.246 (2.855–6.313) | <0.001 | 2.337 (1.449–3.769) | 0.001 |
| CKD | 12.750 (4.613–35.239) | <0.001 | 8.265 (2.868–28.528) | <0.001 |
| Hemoglobin | 1.324 (1.178–1.488) | 0.703 | 1.003 (0.866–1.162) | 0.963 |
| Leukocytosis | 1.179 (1.120–1.242) | <0.001 | 1.130 (1.069–1.195) | <0.001 |
| Platelet | 1.006 (1.002–1.009) | <0.001 | 1.005 (1.000–1.009) | 0.051 |
| Hypoalbuminemia | 5.239 (3.469–7.912) | <0.001 | 2.925 (1.722–4.967) | <0.001 |

found to be 12.8%, which is relatively lower than that reported previously (25–69.2%) [14, 22, 23]. We believe that this finding might be due to referral bias. The incidence of HA found in previous literature is based on data from tertiary care centers; those data were obtained from severe scrub typhus cases [14, 22]. In contrast, our hospital is a secondary care center adjacent to a rural area. Thus, many patients may seek treatment at our hospital while suffering from an early-stage infection that has not yet progressed to a severe stage. Furthermore, some patients showing severe scrub typhus were transferred from our hospital to a tertiary care center. Indeed, the AKI incidence was found to be 23.1% in the present study, whereas Hwang *et al*. reported that the AKI incidence based on RIFLE criteria among scrub typhus cases in a tertiary care center was 35.9% [23].

The occurrence of HA in patients with scrub typhus seems to be multifactorial; poor dietary intake, decreased synthesis of albumin due to hepatic dysfunction, increased catabolism of protein, and proteinuria are all known to be related to hypoalbuminemia in cases of acute infectious disease [11, 14]. The central pathophysiological change associated with scrub typhus is vasculitis and perivasculitis of multiple organs due to the destruction of the endothelial cell lining of small blood vessels and perivascular inflammatory cell infiltration [24–26]. This consequently results in increased vascular permeability with extravascular protein loss. Such a pathomechanism might also contribute to the development of hypoalbuminemia in patients with scrub typhus. Previous studies have demonstrated patients with scrub typhus and HA to have multiple clinical symptoms according to the involved organs [14, 22]; this was also found in the present study. Therefore, it is important to monitor patients closely with scrub typhus with HA.

Among complications in patients with scrub typhus, the development of AKI is clinically important since it is associated with a longer hospital stay and mortality [27, 28]. Old age, comorbidities, hyperbilirubinemia, and biomarkers are known to be risk factors for predicting AKI [9, 10, 29]. Although the association between HA and AKI was not clear in previous studies, a recent study with a larger cohort including our data showed that HA could be a predictor for AKI in scrub typhus [23]. Even though the exact mechanism of AKI development in scrub typhus is not understood, pre-renal factors such as volume depletion, ischemic acute tubular necrosis, and sepsis were regarded as frequent causes of AKI in this population with infectious diseases [12]. In our study, the majority (94%) of patients with AKI either had FENa < 1% or returned to baseline renal function within 72 hours, both of which are suggestive of pre-renal failure. Other patients had fractional sodium excretion >1% and recovered to their baseline renal function over 72 hours. Therefore, although various mechanisms might have contributed to the scrub typhus-associated AKI, we propose that HA is associated with pre-renal failure in scrub typhus-associated AKI.

A negative correlation between serum albumin level and serum creatinine concentration was reported in the study by Kim *et al*., whose participants had a hemorrhagic fever with renal syndrome [30]. However, such a relationship was not demonstrated in scrub typhus-associated AKI. In this study, the serum albumin level on admission was not only a significant predictor of AKI but was also associated with AKI severity based on the RIFLE criteria. Furthermore, patients with HA had a longer hospital stay and required ICU care more frequently compared to patients in NA group. In this study, although the majority of AKI patients recovered their renal function within 72 hours in this study, the eGFR on admission always did not consistent with the lowest value during hospitalization. Therefore, it is especially important to check the serum albumin level to predict the clinical course and prognosis of patients with scrub typhus.

Our study had some limitations. First, this was a retrospective and a single-center study. Second, we did not enroll the data from patients that were treated in the outpatient clinic. Thus, it is possible that we excluded patients with a mild type of scrub typhus. Therefore, a

large prospective randomized controlled study is needed to investigate further whether HA could predict the development of AKI and its severity.

Patients with scrub typhus with HA had various complications including AKI. The incidence of scrub typhus-associated AKI in our study was 23.1%. HA was a significant predictor for scrub typhus-associated AKI and correlated with AKI severity. Therefore, in cases of scrub typhus-associated AKI, checking serum albumin on admission is important to make a therapeutic approach.

## Supporting information

**S1 File. The minimal data set, including the study population characteristics, in this study.** (SAV)

## Author Contributions

**Conceptualization:** In O. Sun.

**Data curation:** Ju Hwan Oh, Ji Hye Lim, A. Young Cho.

**Investigation:** Ju Hwan Oh, Ji Hye Lim, A. Young Cho.

**Supervision:** Kwang Young Lee, In O. Sun.

**Writing – original draft:** Ju Hwan Oh, In O. Sun.

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
