## [Decision Letter · Decision Letter 0]

8 Jan 2021

PONE-D-20-35972

Clinical significance of hypoalbuminemia in patients with scrub typhus complicated by acute kidney injury

PLOS ONE

Dear Dr.In O Sun

Thank you for submitting your manuscript to PLOS ONE. After careful consideration, we feel that it has merit but does not fully meet PLOS ONE’s publication criteria as it currently stands. Therefore, we invite you to submit a revised version of the manuscript that addresses the points raised during the review process.

 After reviewing  the manuscript.I would suggest to please incoprorate the changes as advised by the reviewers in the  article.

We look forward to receiving your revised manuscript.

Kind regards,

Bhagwan Dass, MD

Academic Editor

PLOS ONE

Journal Requirements:

2. Please include additional information regarding the data extraction tool used in the study and ensure that you have provided sufficient details that others could replicate the analyses. For instance, if you developed a data extraction tool as part of this study and it is not under a copyright more restrictive than CC-BY, please include a copy, in both the original language and English, as Supporting Information, or include a citation if it has been published previously.

3. In statistical methods, please refer to any post-hoc corrections to correct for multiple comparisons during your statistical analyses. If these were not performed please justify the reasons. Please refer to our statistical reporting guidelines for assistance (https://journals.plos.org/plosone/s/submission-guidelines.#loc-statistical-reporting).

4. As part of your revision, please complete and submit a copy of the RECORD checklist, a document that aims to improve reporting and reproducibility of observational studies that use routinely-collected data for purposes of post-publication data analysis and reproducibility: (http://record-statement.org). Please include your completed checklist as a Supporting Information file. Note that if your paper is accepted for publication, this checklist will be published as part of your article.

"The authors wish to acknowledge the financial support of the Christian Medical Research

Center, Presbyterian Medical Center, Jeonju, Korea."

"Unfunded study

The authors received no specific funding for this work. "

6. Thank you for submitting the above manuscript to PLOS ONE. During our internal evaluation of the manuscript, we found some minor occurrences of overlapping text with the following previous publication(s), some of which you are an author, which needs to be addressed:

- https://e-sciencecentral.org/articles/pubreader/SC000036783

- https://www.jiac-j.com/article/S1341-321X(13)00033-0/fulltext

- https://synapse.koreamed.org/articles/1128479

We would like to make you aware that copying extracts from previous publications word-for-word, especially outside the methods section, is unacceptable. In addition, the reproduction of text from published reports has implications for the copyright that may apply to the publications.

Please revise the manuscript to quote or rephrase the duplicated text and cite your sources for text outside the methods section. Please note that further consideration is dependent on the submission of a manuscript that addresses these concerns about the overlap in text with published work.

Reviewers' comments:

Reviewer's Responses to Questions

**Comments to the Author**

1. Is the manuscript technically sound, and do the data support the conclusions?

Reviewer #1: Yes

Reviewer #2: Partly

Reviewer #3: Yes

2. Has the statistical analysis been performed appropriately and rigorously? 

Reviewer #1: Yes

Reviewer #2: Yes

Reviewer #3: I Don't Know

3. Have the authors made all data underlying the findings in their manuscript fully available?

Reviewer #1: Yes

Reviewer #2: No

Reviewer #3: Yes

4. Is the manuscript presented in an intelligible fashion and written in standard English?

Reviewer #1: Yes

Reviewer #2: Yes

Reviewer #3: Yes

5. Review Comments to the Author

Reviewer #1: This article poses a clear hypothesis that is fully supported by the data and statistical analysis. The article is clearly written, exhibits good grammar and prose, and the results are clinically relevant. Clinicians may potentially use the findings to direct the level of monitoring various patients receive. Would recommend for publication.

Reviewer #2: The authors present a case series of patients diagnosed with scrub typhus admitted to a medical center in South Korea. They demonstrate that hypoalbuminemia detected on presentation was associated with acute kidney injury, correlated well with severity of the AKI by RIFLE criteria, and was associated with other systemic complications. They further state that multiple clinical and laboratory findings were predictive of AKI by multivariate analysis, which may warrant change to assessment, monitoring, or therapeutic approach.

My assessment of their submission is hampered by duplication of Table 1 in the listed Table 2.

The introduction discusses other publications addressing scrub typhus, its complications, association with renal failure specifically, and association with other complications. The strengths of this analysis by comparison included a larger sample size and use of RIFLE criteria to demonstrate correlation of severity of AKI with severity of hypoalbuminemia. It is interesting that the rate of hypoalbuminemia in this study was lower than others. I would explicitly state whether others included inpatients, outpatients, or both as that is an important limitation of this study.

The section on patient selection should explicitly state whether patients were inpatients, outpatients, or both, and whether they included patients transferred in from another facility. Later in the manuscript, it is stated they excluded outpatients. Also, they exclude patients transferred to other facilities, but it would be helpful to state if it is known why those patients were transferred, especially if it was to a higher level of care.

Elevation of cardiac biomarkers was stated to define myocarditis--this is not accurate and could be stated more accurately as a marker of such. Respiratory complications included ARDS and various imaging findings, but not respiratory failure or mechanical ventilation.

Table 3 should include definition of terms by cutoff values. FENa has a marker for footnote 'a' but no associated footnote and n(%) appear incorrect. If a large number of cases did not have FENa or urinalysis available, I question the value of including that information. Was it assessed whether these cases are representative of the population?

A bigger question is whether HA as an independent biomarker of AKI risk is significant in this setting. It correlated well with AKI and other poor outcomes, but as a negative acute phase reactant, one would expect that to be true. The homeostasis of albumin is affected by numerous factors other than vascular permeability. During acute inflammatory syndromes, synthesis is suppressed and catabolism increased in addition to increased volume of distribution. The "central pathomechanism" comment is incomplete. I would direct to the discussion cited as a reference in Lee, et al. (BMC Infect Dis 2010).

The fact that almost all cases with AKI were present on admission makes HA a marker for a condition that was already present on admission, not a predictor that it will occur. Does HA predict which cases with AKI will progress to a more significant degree of AKI, or which cases that present with HA and normal renal function will progress to AKI? Would it predict which of the 2% of cases will not recover?

Reviewer #3: The authors must be complemented on writing about this very important infectious disease.

I had a few queries:

1. In page 11 could you elaborate further and explain how definiitons for cardiac and respiratory complications were arrived at

2. How was meningoencephalitis identified

3. Under discussion : "The development of HA in

patients with scrub typhus seems to be due to the central pathomechanism of scrub typhus,

which is vasculitis or perivasculitis of multiple organs resulting in increased vascular

permeability". Could you please provide a reference.

4. Further explanation of who were included in the study is warranted. We would like to know what work up was done for patients presenting with fever and a rash. What other causes were ruled out.

5.

6. PLOS authors have the option to publish the peer review history of their article (what does this mean?). If published, this will include your full peer review and any attached files.

Reviewer #1: No

Reviewer #2: No

Reviewer #3: No

---

## [Author Response · Author response to Decision Letter 0]

4 Feb 2021

Thank you for reviewing our manuscript. We have changed the contents one by one according to your comments. Thank you so much.

---

## [Editor Report · Decision Letter 1]

10 Feb 2021

Clinical significance of hypoalbuminemia in patients with scrub typhus complicated by acute kidney injury

PONE-D-20-35972R1

Dear Dr. Sun,

We’re pleased to inform you that your manuscript has been judged scientifically suitable for publication and will be formally accepted for publication once it meets all outstanding technical requirements.

Kind regards,

Bhagwan Dass, MD

Academic Editor

PLOS ONE
---

## [Editor Report · Acceptance letter]

16 Feb 2021

PONE-D-20-35972R1 

Clinical significance of hypoalbuminemia in patients with scrub typhus complicated by acute kidney injury 

Dear Dr. Sun:

I'm pleased to inform you that your manuscript has been deemed suitable for publication in PLOS ONE. Congratulations! Your manuscript is now with our production department. 

Kind regards, 

on behalf of

Dr. Bhagwan Dass 

Academic Editor

PLOS ONE